# Assessment of Calcaneal Spongy Bone Magnetic Resonance Characteristics in Women: A Comparison between Measures Obtained at 0.3 T, 1.5 T, and 3.0 T

**DOI:** 10.3390/diagnostics14101050

**Published:** 2024-05-18

**Authors:** Silvia Capuani, Alessandra Maiuro, Emiliano Giampà, Marco Montuori, Viviana Varrucciu, Gisela E. Hagberg, Vincenzo Vinicola, Sergio Colonna

**Affiliations:** 1CNR-ISC c/o Physics Department, “Sapienza” University of Rome, P.zle Aldo Moro 5, 00185 Rome, Italy; alessandra.maiuro@uniroma1.it (A.M.); marco.montuori@cnr.it (M.M.); 2Neuroimaging Laboratory, Santa Lucia Foundation, IRCCS Rome, Via Ardeatina 309, 00179 Rome, Italy; 3Physics Department, “Sapienza” University of Rome, P.zle Aldo Moro 5, 00185 Rome, Italy; 4Rehabilitation Hospital, Santa Lucia Foundation, IRCCS Rome, Via Ardeatina 309, 00179 Rome, Italy; emiliano.giampa@gmail.com (E.G.); v.vinicola@hsantalucia.it (V.V.); 5Radiology Department, Santa Lucia Foundation, IRCCS Rome, Via Ardeatina 309, 00179 Rome, Italy; viviana.varrucciu@tiscali.it (V.V.); s.colonna@hsantalucia.it (S.C.); 6High Field Magnetic Resonance, Max-Planck-Institute for Biological Cybernetics, 72076 Tübingen, Germany; gisela.hagberg@tuebingen.mpg.de

**Keywords:** MRI, low magnetic field, high magnetic field, spongy bone, bone marrow fat

## Abstract

Background: There is a growing interest in bone tissue MRI and an even greater interest in using low-cost MR scanners. However, the characteristics of bone MRI remain to be fully defined, especially at low field strength. This study aimed to characterize the signal-to-noise ratio (SNR), T_2_, and T_2_* in spongy bone at 0.3 T, 1.5 T, and 3.0 T. Furthermore, relaxation times were characterized as a function of bone-marrow lipid/water ratio content and trabecular bone density. Methods: Thirty-two women in total underwent an MR-imaging investigation of the calcaneus at 0.3 T, 1.5 T, and 3.0 T. MR-spectroscopy was performed at 3.0 T to assess the fat/water ratio. SNR, T_2_, and T_2_* were quantified in distinct calcaneal regions (ST, TC, and CC). ANOVA and Pearson correlation statistics were used. Results: SNR increase depends on the magnetic field strength, acquisition sequence, and calcaneal location. T_2_* was different at 3.0 T and 1.5 T in ST, TC, and CC. Relaxation times decrease as much as the magnetic field strength increases. The significant linear correlation between relaxation times and fat/water found in healthy young is lost in osteoporotic subjects. Conclusion: The results have implications for the possible use of relaxation vs. lipid/water marrow content for bone quality assessment and the development of quantitative MRI diagnostics at low field strength.

## 1. Introduction

Magnetic resonance imaging (MRI) is one of the most promising tools to assess in vivo pathological abnormalities of bone tissues [1,2]. Due to its high sensitivity to water protons, musculoskeletal MRI has found its main applications in investigating soft tissues such as cartilage [3], muscles [4], and tendons [5] to detect injuries, such as fractures or tears to tendons [6], ligaments, or cartilages [7,8,9], and in diagnosing soft tumors [10,11]. MR imaging techniques for studying the musculoskeletal system have undergone strong development over the past 20 years [12], and rapid progress has been made recently related to the structure [13,14,15,16] and quality of spongy bone [17,18,19,20,21,22,23] that is potentially applicable to the clinical evaluation of osteoporosis [24,25,26]. Osteoporosis is characterized by loss of bone mineral in the human skeleton due to metabolic changes primarily affecting the micro-architectural structure of spongy bone. As recommended by the WHO [27], the clinical diagnosis of osteoporosis is currently based on the quantification of bone mineral density (BMD) of those skeletal sites with high trabecular content, such as the spine, proximal femur, and calcaneus [28]. Dual-energy X-ray absorptiometry (DXA) and computed tomography (CT) are the diagnostic tools currently employed in clinical routines for BMD assessment. However, these techniques have some relevant limitations, including the use of ionizing radiation and a low predictive value on patients’ risk of reporting bone fracture (65%). This lack of sensitivity is likely due to the partial information that DXA and CT provide on spongy bone characteristics, assessing exclusively its mineral component. Other components, such as bone marrow, collagen, and proteins, are present in the context of bone tissue and may contribute to determining its resistance to fracture [29,30]. Moreover, osteoporosis is a silent and largely undiagnosed disease [31,32]. People often turn to an osteoporosis diagnostic center when they suffer a fracture. Diagnostic tools that can increase the population undergoing diagnosis would therefore be needed. Ideally, the diagnosis should be radiation-free and low-cost to be accessible to all. Currently, magnetic resonance imaging (MRI) and magnetic resonance spectroscopy (MRS) are proposed as potential radiation-free methods for the diagnosis of osteoporosis. However, the two main MR approaches developed for the evaluation of osteoporosis: (1) MR interferometry [13,15,16,31] and high-resolution MRI [14], and (2) the evaluation of the bone marrow quality with MRS [17,18,19,22,25,26,32], require high-field MR scanners, which have the disadvantage of costing around EUR 1 million per tesla and consequently entail a high cost for the user who needs a diagnosis.

The former approach is based on T_2_*-weighted gradient-echo (GE) imaging of the spongy bone marrow, which exploits the magnetic field inhomogeneities generated by the magnetic susceptibility difference between bone trabecula and biological water [33] and/or on high-resolution MR Imaging, which allows a quantitative and direct morphometric analysis of the three-dimensional structure of cancellous bone. High-resolution MRI is very demanding in terms of field gradient strength and system performance and is hardly applicable in clinical routines. The latter approach requires the acquisition of MR spectra from which extract the fatty acid quantification or the fat/water ratio in the bone marrow. Currently, scanners equipped with magnetic field strengths of 1.5 T and 3.0 T are available for clinical and translational research use.

In recent years, low-cost MR scanners have been developed. They are characterized by low magnetic field strength (less than 0.5 T) and usually are dedicated to specific body parts, such as the extremities. As an example, a single-sided MR with a magnetic field of about 0.3 T has been used to detect skin anomalies [34,35], investigate breast tissue [36], extract tendons and cartilage information indirectly [37,38], or extract geometrical information about trabecular bone microstructures potentially useful for the diagnosis of osteoporosis [39,40]. A compact MRI system for measuring the trabecular bone volume fraction (TBVF) of the calcaneus was developed using a 0.21 T permanent magnet and portable MRI console [41].

Nowadays, it is possible to find detailed information on different MRI parameters related to different cerebral zones and measured as a function of the magnetic field strength [42]. Conversely, few available data for the musculoskeletal system and, in particular, for spongy bone are furnished. The purpose of this study is to characterize the signal-to-noise ratio (SNR), T_2_*, and T_2_ in calcaneus spongy bone at 0.3 T, 1.5 T, and 3.0 T for future investigations related to the diagnosis of osteoporosis and related musculoskeletal dysfunction. Toward this goal, we have investigated the calcaneus because it is the site of the skeleton rich in spongy bone that does not cause claustrophobia problems for patients who underwent an MRI. Moreover, we have evaluated the lipid/water ratio content in the bone marrow at 3.0 T to study the behavior of T_2_ and T_2_* as a function of the water and fat content in young, healthy, and osteoporotic postmenopausal women.

## 2. Materials and Methods

### 2.1. Subject Recruitment

A cohort of thirteen healthy women (H1, mean age, 24 ± 3 years), a cohort of six healthy women (H2, mean age, 27 ± 3 years), and a cohort of thirteen osteoporotic women (OPO, mean age, 62 ± 5 y, mean T-score = −3.1 ± 0.4) of the Caucasian race participated in this study. All subjects were carefully investigated to exclude the presence of any bone disease (apart from osteoporosis for the osteoporotic group), systemic metabolic disorders, and malignancies. Subjects assuming any medication affecting bone mineral homeostasis (e.g., steroids) were also excluded.

This study was approved by the local Ethics Committee (Fondazione Santa Lucia, Rome, Italy, Prot. CE/2023_024), and written informed consent was obtained in all cases before study initiation.

### 2.2. MR Measurements

MRI was performed using three different commercial MR scanners: O-Scan (Esaote, Genova, Italy), Vision, Siemens, and Allegra, Siemens, (Siemens, Erlangen, Germany), operating at 0.31 T, 1.5 T, and 3.0 T, respectively. The cohort H1 of thirteen healthy women was investigated at both 1.5 T and 3.0 T, the cohort OPO of osteoporotic women was investigated at 3.0 T, and the cohort H2 of six healthy women was investigated at 0.3 T.

#### 2.2.1. MR Acquisition at 1.5 T and 3.0 T

Peak gradient amplitudes were 24 mT/m and 40 mT/m for 1.5 and 3.0 T, respectively, while slew rates were 180 mT/m/ms for 1.5 T and 400 mT/m/ms for 3.0 T systems. A circular polarized volume head-coil for radiofrequency (RF) transmission and reception was used in both cases. Subjects were placed in a supine position on the imaging table with the right foot inside the head RF coil. Thus, sagittal view images obtained on the same slices (5 mm thickness) and by using the same foot position were acquired for every volunteer. Specifically, sagittal images were obtained parallel to the long axis of the calcaneus. FLASH (fast low-angle shot) and MCSE (multi-contrast spin-echo) images at various TEs were collected to evaluate T_2_* and T_2_, respectively.

The imaging parameters used for MCSE images at both 1.5 T and 3.0 T were as follows: echo time (TE) = 20, 45, 80, 120 ms; bandwidth (BW) = 130 Hz/pixel; square field of view (FOV) = 192 mm; matrix = 256 × 256 pixels; and resolution = 0.75 × 0.75 × 5 mm^3^. For FLASH images, the following parameters at both 1.5 T and 3.0 T were used: TE = 5, 7, 10, 20 ms; BW = 260 Hz/pixel; square FOV = 192 mm; matrix = 128 × 128 pixels; and resolution = 1.5 × 1.5 × 5 mm^3^. Finally, TR = 3000 ms, number of signals (NSs) = 1, and slice thickness (STK) equal to 5 mm were used in all experiments.

Table 1 summarizes the MR acquisition protocol at 1.5 T and 3.0 T used for any studied subject.

We also performed a FLASH T_2_*–T1-weighted image at 3.0 T using square FOV = 180 mm, matrix = 512 × 512, TE = 10 ms, TR = 600 ms, and BW = 160 Hz/pixel as a reference to discriminate the calcaneus zones ST, TC, CC characterized by different trabecular bone density. In Figure 1, an example of this acquisition is reported with the selected subtalar (ST), tuber calcanei (TC), and cavum calcanei (CC).

ROIs. In all the MRI sequences, no chemical pre-saturation pulses were used for either fat or water protons.

#### 2.2.2. MR Acquisition at 0.3 T

A permanent magnet (constituted by multiple magnets) of 0.31 T, peak gradient amplitudes of 20 mT/m, and slew rates of 100 mT/m/ms was used with a dual-phase array knee coil for radiofrequency (RF) transmission and reception. Subjects placed their right foot inside the RF coil. Thus, sagittal view images of 7 mm thickness were acquired for every volunteer. Specifically, GE (gradient-echo) at TE = 10, 14, and 16 ms and FSE (fast spin-echo) images at various TEs from 25 to 200 ms were collected to evaluate T_2_* and T_2_, respectively. The in-plane resolution was =0.55 × 0.55 mm^2^ for both T_2_ and T_2_* weighted images. Table 2 summarizes the MR acquisition protocol at 0.3 T used for any studied subject.

### 2.3. SNR, T_2_, and T_2_* Measurements

As reported in the Introduction, T_2_* and T_2_ measurements of spongy bone are indicated in the literature as promising parameters to develop an NMR approach to diagnose osteoporosis. To test the potentiality of this method in detecting variation in trabecular bone density, as occurs in osteoporosis, we focused our attention on three different calcaneal sites characterized by different trabecular bone density: the subtalar (ST), the tuber calcanei (TC) and the cavum calcanei (CC) as represented in Figure 1. The ST region is characterized by the highest trabecular density. Trabecular density progressively decreases when moving from TC to CC regions. The CC region is characterized by the lowest and isotropic trabecular density [43]. To evaluate SNR, the most common TE values used in radiological imaging of spongy bone were chosen for FLASH and MCSE images. At 1.5 T and 3.0 T, the SNR was measured from FLASH and MCSE images at TE = 5 ms and 45 ms, respectively, as the ratio of the mean signal, measured in each of the three regions of interest (ROI), and the mean value of the background noise (measured in a region of no signal). For consistency, ROIs were placed at an identical position on each image by the same operator (shown in Figure 1b), selecting the slice of the center of the calcaneus.

The SNR at 0.3 T was calculated in the whole calcaneal area and then compared with that acquired at 1.5 T, selecting GE and FSE images (at 0.3 T) and FLASH and MCSE images (at 1.5 T) using TE = 10 ms and TE = 50 ms for gradient-echo and spin-echo images. We had to use TEs different from those selected for the comparison between 1.5 T and 3.0 T due to less flexibility of the low-field scanner, which, for example, only allows TE = 10, 14, and 16 ms to be selected for the GE acquisition sequence.

T_2_* and T_2_ from gradient-echo and spin-echo images, respectively, were obtained by performing a mono-exponential fit of the mean intensities of every selected calcaneal ROI and at the whole calcaneal area at the different TEs.

The equation (Equation (1))
(1)$$S\left(TE\right)=S\left(0\right)\exp\left(\frac{-TE}{T_{2}}\right)+c$$
was fitted to signal decay data. The term S(0) is the signal at TE = 0, which represents the equilibrium magnetization M0, and the T_2_ term represents T_2_* and T_2_ for gradient-echo and spin-echo experiments, respectively. The term c is a constant that takes into account the noise level.

### 2.4. Single-Voxel Spectroscopy at 3.0 T

As the recent literature suggests changes in bone marrow fat content with the development of osteoporosis, we evaluated the dependence of T_2_ and T_2_* parameters on fat/water ratio. Towards this goal, single-voxel spectroscopy (SVS) was performed at 3.0 T with point-resolved spectroscopy (PRESS) sequence and with TE = 22 ms, TR = 5 s, and NS = 32 to obtain bone marrow proton spectra. The voxel size of 15 × 15 × 15 mm^3^ was positioned in the center of the calcaneus.

### 2.5. Data Analysis

#### 2.5.1. SNR, T_2_, and T_2_* Evaluation

A modified Levenberg–Marquardt nonlinear regression fit-type function (using MATHLAB software) was used to obtain relaxation time values from all cohorts of subjects. SNR and relaxation times were averaged across all subjects, and their standard errors were calculated using the propagation of errors. *p*-values were calculated using a paired Student’s *t*-test.

The mean percentage of SNR gain at 3 T compared to 1.5 T was calculated for each ROI using the following equation (Equation (2)):(2)$$\bar{SNR}g(\%)=\frac{{\bar{\left(SNR\right)}}_{3T}-{\bar{\left(SNR\right)}}_{1.5T}}{{\bar{\left(SNR\right)}}_{1.5T}}*100$$
where ${\bar{\left(SNR\right)}}_{3.0T}$ is the mean value of SNR at 3.0 T magnetic field, and ${\bar{\left(SNR\right)}}_{1.5T}$ is the mean value of SNR at 1.5 T. A two-way ANOVA was used to investigate the effect of the magnetic field strength on the T_2_* and T_2_ values of the three considered calcaneal regions.

Using Equation (2), the mean percentage of SNR gain at 1.5 T compared to 0.3 T was calculated considering the whole calcaneus. Moreover, as the slice thickness of images is 7 mm and 5 mm for investigations at 0.3 T and 1.5 T, respectively, the SNR at 0.3 T was multiplied for 5/7.

#### 2.5.2. Bone Marrow Lipid to Water Concentration Ratio

As mentioned above, ^1^H-MR spectroscopy was used to evaluate the dependence of the relaxation times as a function of the fat-to-water concentration ratio. Row data of spectra acquired from each subject at 3.0 T were analyzed using LCModel (SPTYPE 6) [44]. Methylene (CH_2_) and methyl (CH_3_) peak areas (at 1.3 ppm, 1.6 ppm, and 0.9 ppm) and water (H_2_O) (at 4.7 ppm) peak areas were calculated for each spectrum. Then, the CH_2_+CH_3_/H_2_O ratios were derived (provided by LCModel as L16 + L13 + L09 resonance normalized to water) and correlated with T_2_ and T_2_* values using Pearson’s correlation coefficient.

Other than the bone marrow lipid/water quantity is also used the bone-marrow fat content percentage C, which is equal to
(3)$$C(\%)=\frac{lipid}{lipid+water}\times 100$$

## 3. Results

Examples of gradient-echo (T_2_*-weighted) acquisitions obtained at 0.3 T, 1.5 T, and 3.0 T are reported in Figure 2.

### 3.1. SNR and Relaxation Times

#### 3.1.1. Results at 1.5 T and 3.0 T

The results reported in Table 3 show that the mean SNR gain ($\bar{SNR}g$) at 3.0 T compared to 1.5 T is different for the three calcaneal regions in both FLASH and MCSE images. The lowest was found in the subtalar region, the highest in the cavum calcanei, while there was an intermediate value in the tuber calcanei. The $\bar{SNR}g$ change was more prominent in MCSE than in FLASH images. Preliminary assessments of SNR gain were performed on a homogeneous phantom, reporting values of about 100% in both MCSE and FLASH images. The results can be better understood by comparing the values reported in Table 3 with the transverse relaxation times results reported in Table 4 and Table 5. T_2_* values were also different among the considered calcaneal regions, with the highest values in the cavum calcanei and the lowest in the subtalar region. Intermediate values were found in the tuber calcanei. Furthermore, T_2_* and T_2_ at 3.0 T were significantly lower than the corresponding values measured at 1.5 T (Table 5 and Table 6). However, the percentage decrease was significantly higher for T_2_* than for T_2_. These results are statistically significant, as indicated by the calculated *p*-values. A two-way ANOVA analysis shows that the mean T_2_ values are significantly different at the two field strengths (*p* = 0.0062), but there is no difference in T_2_ between the three trabecular regions (*p* = 0.69) and no interaction effect (*p* = 0.914) (Table 5). Conversely, the mean T_2_* values differed significantly between fields (*p* = 0.0001) and between trabecular bone regions (*p* < 0.0001); however, there was no interaction effect (*p* = 0.998), indicating that the T_2_* values decreased similarly in all regions when the magnetic field strength increased.

#### 3.1.2. Results at 0.3 T and 1.5 T

Results reported in Table 6 show the mean SNR gain ($\bar{SNR}g$) at 1.5 T compared to 0.3 T is different obtained in both gradient-echo (i.e., FLASH and GE) and spin-echo (i.e., MCSE and FSE) images and considering the whole calcaneus. Moreover, T_2_* and T_2_ values obtained at 0.3 T and 1.5 T in the whole calcaneus are displayed in Table 7.

#### 3.1.3. Dependence of Relaxation Times on Lipid/Water Ratio

An example of a bone marrow NMR spectrum obtained in the calcaneus at 3 T is displayed in Figure 3, with the images for the voxel localization and the fit (in red) obtained using LCModel to extract resonance quantification.

In Figure 4, relaxation times T_2_ and T_2_* obtained at 3.0 T as a function of the calculated L16 + L13 + L09-to-water peak area ratios are reported for each of the three calcaneus sites: subtalar (a, d), tuber calcanei (b, e), and cavum calcanei regions (c, f). The graphs in Figure 4 show that both spongy-bone T_2_ and T_2_* strongly depend on the lipid-to-water ratio present in the bone marrow. It is important to note that even in a very selective healthy women group (small age range, same race), a wide variability of bone marrow fat content was observed, and a high significant (*p* < 0.001) linear correlation was found between both T_2_ and T_2_* and the lipid/water in healthy young women. On the other hand, in Figure 5, relaxation times T_2_* obtained at 3.0 T as a function of the lipid/water (or bone marrow fat content percentage) in the whole calcaneus of the healthy and osteoporotic group are displayed. Interestingly, the significant linear correlation observed in young, healthy subjects is completely lost when the osteoporotic group is examined.

## 4. Discussion

In NMR theory, the signal is proportional to the square of the static magnetic field strength, and the noise is proportional to the static magnetic field strength. Therefore, a 3.0 T MR system can theoretically achieve two times the SNR of a 1.5 T system (i.e., the increase in SNR is 100%), and a 1.5 T MR system can theoretically achieve five times the SNR of a 0.3 T MRI system (with an SNR gain of 400%). The results regarding the SNR gain reported in this work indicate lower values compared to those predicted by theory, suggesting that many other factors affect the gain in SNR when the intensity of the magnetic field increases, as already observed by some authors [45,46,47]. The SNR also depends on the properties of the object to be imaged and the scanning acquisition and instrumentation.

In this paper, we have evaluated the SNR gain at 3.0 T compared to 1.5 T and at 1.5 T compared to 0.3 T, considering the calcaneal spongy bone. Spongy bone consists of a three-dimensional network in which bone marrow, mainly containing water and fat, is dispersed in the interstitial spaces. The susceptibility mismatch between the solid matrix (composed of trabecular bone network) and the interstitial liquid (composed of bone marrow) causes an induced local magnetic field that generates inhomogeneities of the static magnetic field [48,49,50], thus generating the so-called internal gradient Gi [16,33]. The effective transverse relaxation time T_2_* is sensitive to the difference in magnetic susceptibility between trabecular bone and bone marrow. The dephasing of the transverse magnetization due to susceptibility differences produces a T_2_* shortening. An increase in trabecular spacing, for instance, induced by osteoporosis, reduces the spatial field inhomogeneity and prolongs T_2_* [51]. This effect is clearly visible in the images shown in Figure 2 and in Table 3, Table 4, Table 6, and Table 7. As bone density increases, T_2_* decreases, and the gain in SNR decreases at 3.0 T compared to 1.5 T. In fact, in our study, the lower T_2_* and lower gain in SNR at 3.0 T compared to 1.5 T is obtained by analyzing the ST area of the calcaneus. This shows a particular sensitivity of T_2_* to the density of the spongy bone, which, in fact, is already the subject of studies for a possible diagnosis of osteoporosis by NMR [13,15,16,31,33]. The effect of the local field inhomogeneity generated by the magnetic susceptibility difference between bone trabeculae and bone marrow depends on the magnetic field strength. As the magnetic susceptibility difference between bone and water is about 1 ppm, much greater than the usual magnetic susceptibility differences found in cerebral tissues [52], the magnetic susceptibility mismatch effect is more pronounced in spongy bone than in other cerebral tissues. As a consequence, no information found in the literature relating to T_2_* or SNR obtained on brain tissues at different magnetic field strengths [53,54] can be used for spongy bone and/or musculoskeletal tissues. In this work, we found a T_2_* value obtained at 0.3 T much greater than those obtained at 1.5 T and 3.0 T (Table 4 and Table 7) with a consequent smaller increase in SNR expected at 1.5 T compared to 0.3 T, in reference to that expected from NMR theory. Also, the small chemical shift difference between fat and water resonance that is equal to about 44 Hz at 0.3 T compared to 440 Hz at 3.0 T contributes to a better image quality at 0.3 T (Figure 2) [55].

On the other hand, the transverse relaxation time T_2_ obtained by a spin-echo (SE) sequence [56] is less sensitive to differences in magnetic susceptibility. This is because the 180° radiofrequency pulse in the SE sequence refocuses all the static magnetic field inhomogeneity [56]. Regarding the non-static magnetic field inhomogeneity, in spongy bone, the diffusion of bone marrow molecules in the local magnetic field gradient Gi becomes an important factor [57]; molecules interchange their positions, resulting in a small phase difference between their nuclear magnetic moments, thus generating an irreversible signal loss [58,59,60]. As a consequence, a dependence of T_2_ by Gi and diffusion D is observable in spongy bone [16,33]. To reduce diffusion and Gi effects on T_2_ values, the echo train acquisition sequence has been developed [61,62]. In this work, the images obtained using the 0.3 T scanner were acquired using FSE for the T_2_ contrast. FSE sequences, due to the use of a 180° echo train (the number of which is indicated by the parameter ETL), show the advantage of a strong reduction in susceptibility artifacts, field inhomogeneity, and acquisition time [55,62]. For this reason, the experimental SNR gain at 1.5 T compared to 0.3 T in cancellous bone is about half of what was expected (experimental $\bar{SNR}g(\%)$ = 260 and theoretical $\bar{SNR}g(\%)$ = 400).

The T_2_ relaxation time of the calcaneus estimated at 1.5 T and 3.0 T with conventional spin-echo acquisition sequences is approximately 50 ms, while that estimated at 0.3 T is approximately 80 ms. Since the T_2_ should not change as a function of the magnetic field strength, the difference between the values is due to the effect of the Gi and molecular diffusion coupling, which contributes to decreasing the value of T_2_ at 1.5 T and 3.0 T compared to that calculated at 0.3 T.

Results related to relaxation times as a function of bone marrow fat content obtained in young, healthy women underline the high sensitivity of relaxation times to bone marrow characteristics. It is well known that the fat fraction percentage in bone marrow is site and age-dependent [63,64]. Fat bone marrow content increases with age, and it is higher in calcaneal than in vertebral spongy bone, whose bone marrow is characterized by a higher content of water [21,63,64]. This work underlines a wide variability of lipids/water ratios and relaxation times values, even in the bone marrow of young healthy women with a very narrow age range (21–27 years). This suggests that bone marrow includes information about an individual. For example, correlations between the quality of bone marrow and nutrition are recently being studied [65] or bone marrow quality concerning physical activity [66]. Moreover, the loss of the significant linear correlation between T_2_* values and marrow fat content percentage when the osteoporotic group is investigated underlines changes in bone marrow components compared to normal healthy bone marrow. In this perspective, the measurement of T_2_ and T_2_* relaxation times, regardless of image resolution or magnetic field intensity, could be used for personalized diagnostics. Towards this goal, low-cost dedicated scanners should be optimized for the quantification of the relaxation times and other quantitative MRI parameters, such as the molecular diffusion coefficient D, another parameter very sensitive to bone marrow changes [16].

This work has limitations. First of all, we used a small group of subjects. Furthermore, in Figure 5, the results obtained in healthy and young women (age range 21–27 years) are compared with those obtained in osteoporotic women aged between 57 and 67 years, not considering the effects of normal aging. Furthermore, T_2_* at the low magnetic field was estimated using only three images at three echo times of values much smaller than the estimated T_2_*.

The development of MRI protocols with low-cost, low-field scanners can help greatly reduce the cost of diagnostics so that they can be available to a greater number of women. It would also be possible to make this technology accessible to elderly men who are affected by osteoporosis at a ratio of one to five compared to women. Furthermore, since the technology is radiation-free, it could also be useful for identifying and monitoring pathologies related to the quality of bone marrow and bones in children and adolescents. However, low-cost instrumentation hardware and software should allow quantification of NMR parameters such as T1 and T_2_ relaxation times and diffusion coefficient of musculoskeletal tissue. Therefore, it should be possible to change the value of the echo time (TE) from very small values to values comparable with tissue T_2_ or T_2_* to better evaluate these parameters. In this work, for example, we reported that the Esaote O-SCAN at 0.3 T allowed the selection of only three echo times to evaluate the T_2_* parameter (see Table 2). This is related to the fact that low-cost clinical scanners are mainly made to obtain images adequately weighted in some NMR parameters to better visualize and contrast the different tissues rather than quantifying NMR parameters that are potential biomarkers of musculoskeletal tissue pathologies.

## 5. Conclusions

This work highlights that changes in MRI characteristics of spongy bone tissue due to variations in magnetic field intensity differ from those widely reported in the literature on brain tissue. Furthermore, this study highlights a specific and sensitive detection of bone marrow quality (in terms of lipids/water) using transverse relaxation times, which would be desirable to develop and test with low-cost scanners [67,68].

This work suggests the optimization of a low-cost dedicated MRI scanner to be used to develop new protocols based on the quantification of MRI relaxation and diffusion parameters for the diagnosis of osteoporosis. This is because the SNR decrease compared to conventional 1.5 T and 3.0 T is not as dramatic as predicted by the theory, especially when spongy bone is analyzed. This is mainly due to the reduced magnetic susceptibility differences between tissues, the reduced chemical shift between water and fat at lower magnetic field strengths, and the development of acquisition sequences that can noticeably improve image quality. Moreover, the quantification of relaxation times and diffusion MRI parameters in spongy bone sites, such as calcaneus, does not require high-resolution images but requires the possibility to change the acquisition parameters to optimize the quantification of MRI parameters, which may serve as potential biomarkers of pathologies.

## Figures and Tables

**Figure 1 diagnostics-14-01050-f001:**
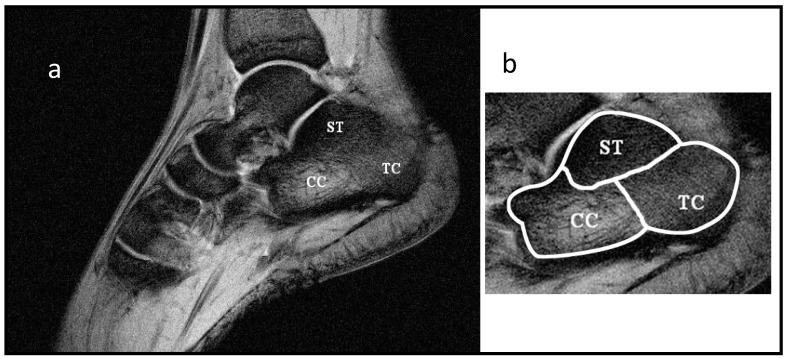
(**a**) T_2_*–T1-weighted image obtained at 3.0 T. (**b**) Selection of ROIs that correspond to the three calcaneus areas of interest: ST, subtalar region, TC, tuber calcanei region, CC, cavum calcanei region. Image resolution was 0.35 × 0.35 × 5 mm^3^.

**Figure 2 diagnostics-14-01050-f002:**
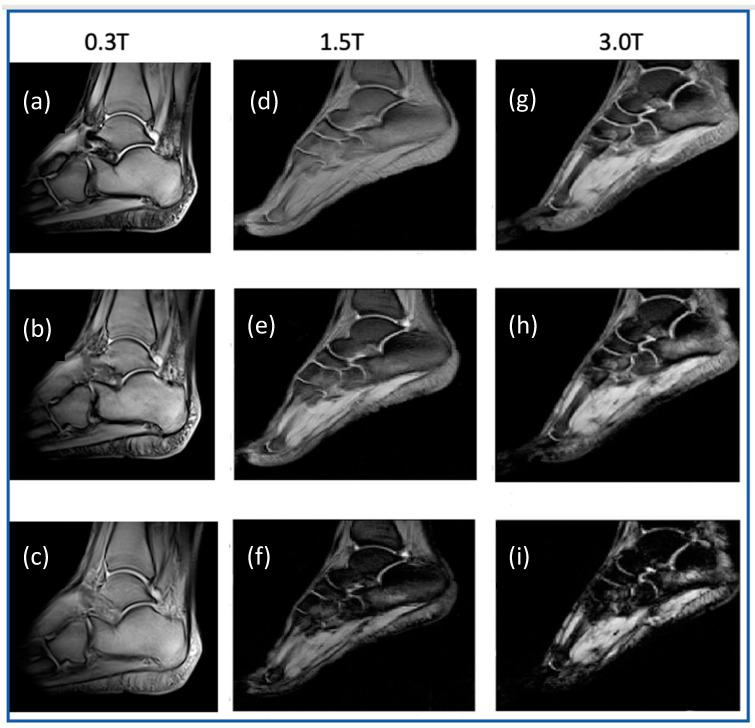
T_2_*-weighted images obtained at 0.3 T, 1.5 T, and 3.0 T in young women. Specifically, images (**a**–**c**) were obtained using a low-cost 0.3 T scanner dedicated to the extremities using TE = 10 ms, 14 ms, and 16 ms, respectively. Images (**d**–**f**) and (**g**–**i**) are obtained at 1.5 T whole-body scanner and 3.0 T head dedicated scanner using TE = 5, 10, and 20 ms. Images obtained at 1.5 T and 3.0 T are of the same volunteer. Image resolution is 0.55 × 0.55 × 7 mm^3^ for those obtained at 0.3 T and 1.5 × 1.5 × 5 mm^3^ for those obtained at 1.5 T and 3.0 T. The different image contrasts are due to the magnetic susceptibility differences between tissues that increase in parallel to the magnetic field strength increase.

**Figure 3 diagnostics-14-01050-f003:**
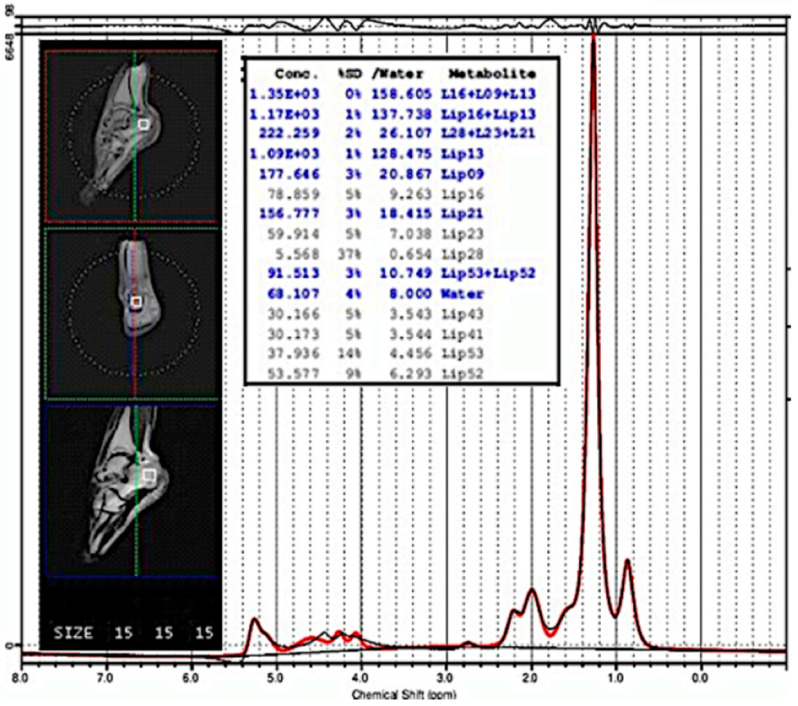
An example of a bone marrow NMR spectrum obtained in calcaneus at 3 T using SVS PRESS (TE/TR = 22/5000 ms) together with the images for the voxel localization in calcaneus. The LC-Model [44] fit to spectrum data (in black) is reported in red. The extract resonance quantifications and their standard deviation (SD) are displayed in the insert.

**Figure 4 diagnostics-14-01050-f004:**
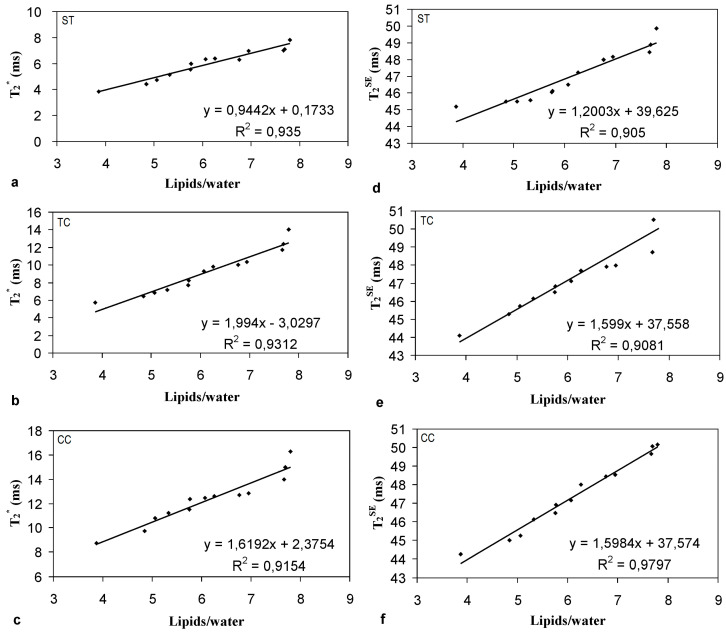
Thirteen young volunteers’ transverse relaxation times in subtalar (**a**,**d**), tuber calcanei (**b**,**e**), and cavum calcanei regions (**c**,**f**) versus fat-to-water concentration ratio. Their linear correlations (R^2^ coefficients) and their functional linear dependency y(x) are also shown.

**Figure 5 diagnostics-14-01050-f005:**
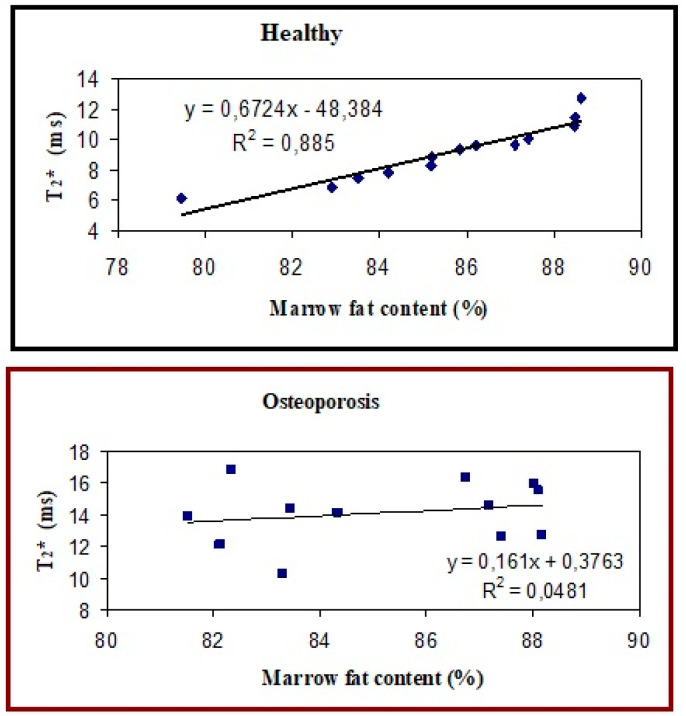
Comparison between the T_2_* dependence on lipid/water in the healthy young and osteoporotic group obtained in the whole calcaneus at 3.0 T. The significant linear correlation in the healthy group is lost when osteoporotic subjects are investigated.

**Table 1 diagnostics-14-01050-t001:** MR acquisition sequences and parameters used to acquire data at 1.5 T and 3.0 T.

	T_2_* Measurements	T_2_ Measurements
Acquisition sequence	FLASH	MCSE
TR (ms)	3000	3000
TE (ms)	5	20
	7	45
	10	80
	20	120
Flip angle	30°	90°, 180°
FOV (mm^2^)	192 × 192	192 × 192
Bandwidth (Hz/pixel)	260	130
Matrix (pixels)	128 × 128	256 × 256
Slice thickness (mm)	5	5
Number of slices	10	10
Slice gap	0	0
Number of signals acquired	1	1
Acquisition time	10 min	20 min

TR, repetition time; TE, echo time; FOV, field of view; FLASH, fast low-angle shot; MCSE, multi-contrast spin echo.

**Table 2 diagnostics-14-01050-t002:** MR acquisition sequences and parameters used to acquire data at 0.3 T.

	T_2_* Measurement	T_2_ Measurement
Acquisition sequence	GE	FSE
TR (ms)	2500	2500
TE (ms)	10	25
	14	50
	16	75
		100
		125
		150
		175
		200
ETL	1	10
Flip angle	30°	90°, 180°
FOV (mm^2^)	140 × 140	140 × 140
Matrix (pixels)	256 × 256	256 × 256
Slice thickness (mm)	7	7
Number of slices	10	10
Slice gap	0	0
Number of signals acquired	1	1
Acquisition time	4 min	10 min

TR, repetition time; TE, echo time; ETL, echo-train length; FOV, field of view; GE, gradient echo; FSE, fast spin echo.

**Table 3 diagnostics-14-01050-t003:** Mean SNR percentage gains, $\bar{SNR}g(\%)$, with their standard errors obtained at 3.0 T compared to 1.5 T in the three calcaneal sites in FLASH and MCSE images.

Acquisition Sequence	ST	TC	CC
FLASH (TE = 5 ms)	29 ± 5	38 ± 5	44 ± 5
MCSE (TE = 45 ms)	88 ± 6	92 ± 6	95 ± 7

ST, subtalar region; TC, tuber calcanei region; CC, cavum calcanei region; FLASH, fast low-angle shot; MCSE, multi-contrast spin echo.

**Table 4 diagnostics-14-01050-t004:** Mean T_2_* (ms) with their standard errors obtained at 1.5 T and 3.0 T in the three calcaneal sites (ST, TC, and CC) and percent decreases relative to 1.5 T.

	ST	TC	CC	P (Two-Way ANOVA)
1.5 T	7.9 ± 0.4	11.2 ± 0.9	14.3 ± 1.6	0.0001
3.0 T	6.0 ± 0.3	9.1 ± 1.8	12.3 ± 1.8	
Decrease (%)	25	18	14	
*p*	0.0001	0.0001	0.0005	
P (two-way ANOVA)		<0.0001		

ST, subtalar region; TC, tuber calcanei region; CC, cavum calcanei region.

**Table 5 diagnostics-14-01050-t005:** Mean T_2_ (ms) with their standard errors obtained at 1.5 T and 3.0 T in the three calcaneal sites (ST, TC, and CC) and percent decreases relative to 1.5 T.

	ST	TC	CC	P (Two-Way ANOVA)
1.5 T	48.3 ± 1.0	48.3 ± 0.9	48.8 ± 1.2	0.0062
3.0 T	47.0 ± 1.7	47.4 ± 1.8	47.4 ± 1.8	
Decrease (%)	2.7	1.9	2.6	
*p*	0.0001	0.0001	0.0005	
P (two-way ANOVA)		0.69		

ST, subtalar region; TC, tuber calcanei region; CC, cavum calcanei region.

**Table 6 diagnostics-14-01050-t006:** Mean SNR percentage gains, $\bar{SNR}g(\%)$, with their standard errors obtained at 1.5 T compared to 0.3 T in the whole calcaneus for gradient-echo and spin-echo images.

Acquisition Sequence	$\bar{SNR}g(\%)$
FLASH/GE (TE = 10 ms)	200 ± 14
MCSE/FSE (TE = 50 ms)	260 ± 20

FLASH, fast low-angle shot; GE, gradient echo; MCSE, multi-contrast spin echo; FSE, fast spin echo.

**Table 7 diagnostics-14-01050-t007:** Mean T_2_* (ms) values with their standard errors obtained at 1.5 T compared to 0.3 T in the whole calcaneus.

	T_2_*	T_2_
0.3 T	70 ± 9	83 ± 6
1.5 T	13 ± 2	50 ± 2
% Decrease	81	40

## Data Availability

Data will be made available upon request by writing to Silvia Capuani (silvia.capuani@isc.cnr.it).

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
