# Peer review of "Assessment of Calcaneal Spongy Bone Magnetic Resonance Characteristics in Women: A Comparison between Measures Obtained at 0.3 T, 1.5 T, and 3.0 T"

_diagnostics, 2024, doi:10.3390/diagnostics14101050_

Round 1
Reviewer 1 Report
Comments and Suggestions for Authors
This study reveals a comparison between various magnetic fields (0.3T, 1.5T, and 3.0T) using magnetic resonance imaging (MRI) on the calcaneal spongy bone in women. The focus is on SNR (signal-to-noise ratio), T2 and T2* relaxation times with respect to bone marrow lipid/water ratio as well as the trabecular density of bones. Therefore, this study fills an important knowledge vacuum regarding the quality of bones while also suggesting possible ways to improve the diagnosis of osteoporosis using MRI technology. Overall, the manuscript is well-written and structured, I only have a few recommendations to improve the quality of the manuscript.
Recommendations:
1. While the study focuses on women, discussing the potential applicability of the findings to broader populations, including men and children, could enhance the relevance of the work. Also, exploring whether these MRI techniques and findings could be generalised to other bones prone to osteoporotic fractures and other orthopaedic conditions would be insightful.
2. The manufacturer and model, settings, calibration procedures – all of these should be more detailed. This also involves indicating pulse sequences and image analysis software specifics.
3. The authors should address why such a sample size was chosen and it should also include any power analysis that may have been done to find out whether the research was powered enough to detect significant differences. Additionally, making use of a wider age group or providing more varied demographic information could improve the robustness of the findings. This should be addressed as a limitation if insufficient data can be provided.
4. It would be helpful to discuss the limitations of the study in more detail, such as the impact that different magnetic field strengths used in MRI scans may have on findings, what might be missed by only testing women, and restrictions imposed by the machine’s design. This will ensure a fair assessment of the work done while also outlining possibilities for further investigations.
5. To elaborate on the clinical relevance of their findings, the authors could consider discussing how these MRI parameters might be incorporated into regular clinical practice or whether they are more effective or costlier than existing methods for screening osteoporosis.
Comments on the Quality of English LanguageMinor editing is required.
Author Response
- Q. While the study focuses on women, discussing the potential applicability of the findings to broader populations, including men and children, could enhance the relevance of the work. Also, exploring whether these MRI techniques and findings could be generalised to other bones prone to osteoporotic fractures and other orthopaedic conditions would be insightful.
R. We thank the reviewer for the suggestion. We have now inserted at the end of the discussion section, a new paragraph:
"The development of MRI protocols with low-cost, low-field scanners can help greatly reduce the cost of diagnostics so that they can be available to a greater number of women. It would also be possible to make this technology accessible to elderly men, who are affected by osteoporosis in a ratio of 1 to 5 compared to women. Furthermore, since the technology is radiation-free, it could also be useful for identifying and monitoring pathologies related to the quality of bone marrow and bones in children and adolescents." (Lines 438-443).
- Q. The manufacturer and model, settings, calibration procedures – all of these should be more detailed. This also involves indicating pulse sequences and image analysis software specifics.
R. We have already indicated the manufacturer and model of scanners used for the work:
For the low-field investigation we have used a O-Scan Esaote, Genova, Italy,
For the 1.5T investigation, we used a Vision, Siemens Erlangen, Germany
and for 3.0T we used an Allegra, Siemens, Erlangen, Germany) (Lines 114-115).
The acquisition sequences used are reported in Tables 1 and 2 with the acquisition parameters. We used MathLab software to process images and to perform function fit to data, as reported in lines 430-431.
- The authors should address why such a sample size was chosen and it should also include any power analysis that may have been done to find out whether the research was powered enough to detect significant differences. Additionally, making use of a wider age group or providing more varied demographic information could improve the robustness of the findings. This should be addressed as a limitation if insufficient data can be provided.
R. The sample of young women used to obtain the SNR gain and the linear correlation between relaxation times and the amount of fat and water was chosen with a very small age range, to be as homogeneous as possible. Furthermore, the volunteers were asked to undertake a demanding task, as each of them underwent T2, T2* and SNR measurements at both 1.5T and 3T. Furthermore, at 3.0T the same volunteers also underwent localized spectroscopy. So it was not easy to find a group of women of a narrow age range willing to participate in this study. We had planned 15 but then for various reasons (for example cancellations) we investigated 13. Regarding menopausal women with osteoporosis, we turned to the osteoporosis center of the Santa Lucia Foundation in Rome where we carried out the project. Since we had already analyzed 13 young, healthy women, we recruited the same number of women with osteoporosis.
This work was the initial phase of what we are carrying out now, considering female subjects with an age range of 40-80 y. We therefore included the limitations related to the woman's age range in the discussion section:
"This work has limitations. First of all, we used a small group of subjects. Furthermore, in Figure 5 the results obtained in healthy and young women (age range 21-27 years) are compared with those obtained in osteoporotic women aged between 57 and 67 years, not considering the effects of normal aging. Furthermore, T2* at the low magnetic field was estimated using only three images at three echo times of values much smaller than the estimated T2*." (Lines 433-437)
- Q. It would be helpful to discuss the limitations of the study in more detail, such as the impact that different magnetic field strengths used in MRI scans may have on findings, what might be missed by only testing women, and restrictions imposed by the machine’s design. This will ensure a fair assessment of the work done while also outlining possibilities for further investigations.
R. We thank you very much the Reviewer for this question. We think to have extensively discussed the impact that the different magnetic field strengths used in MRI scans can have on the results, having compared the results at 3.0T, 1.5T and 0.3T. The work is dedicated to postmenopausal osteoporosis and therefore we focused on investigations for women. Certainly, in the case of male osteoporosis, we will have different parameters because the spongy bone of men is more compact than that of women (reduced T2*) and certainly the variation in the relative quantity of fatty acids in the marrow will have different behavior, but this is beyond the scope of our work, which is focused on the spongy bone characteristics at different magnetic field strength and to the diagnosis of female osteoporosis.
Regarding the design of the machine, we have added a comment at the end of the discussion section which only now, thanks to the Reviewer's suggestion, do we consider essential: "
"However, low-cost instrumentation hardware and software should allow quantification of NMR parameters such as T1 and T2 relaxation times and diffusion coefficient of musculoskeletal tissue. Therefore, it should be possible to change the value of the echo time (TE) from very small values to values comparable with tissue T2 or T2* to better evaluate these parameters. In this work, for example, we reported that the Esaote O-SCAN at 0.3T, allowed the selection of only three echo times to evaluate the T2* parameter (see Table 2). This is related to the fact that low-cost clinical scanners are mainly made to obtain images adequately weighted in some NMR parameters to better visualize and contrast the different tissues, rather than quantifying NMR parameters that are potential biomarkers of musculoskeletal tissue pathologies." (Lines 443-451)
- Q. To elaborate on the clinical relevance of their findings, the authors could consider discussing how these MRI parameters might be incorporated into regular clinical practice or whether they are more effective or costlier than existing methods for screening osteoporosis.
R. In our opinion, considering the present work, it is premature to discuss how MRI parameters could be incorporated into clinical practice. We are also now investigating their effectiveness with a new project.
All changes are highlighted in yellow on the manuscript.
Reviewer 2 Report
Comments and Suggestions for Authors
The study Assessment of Calcaneal Spongy Bone MR Characteristics in Women: A Comparison Between Measures Obtained at 0.3T, 1.5T, and 3.0T investigates the signal-to-noise ratio (SNR), T2, and T2* relaxation times of bone tissue MRI across different magnetic field strengths. Utilizing MRI on thirty-two women's calcaneus at 0.3T, 1.5T, and 3.0T, the research explores how these metrics vary with changes in field strength and how they relate to bone-marrow lipid/water ratio and trabecular-bone density. Key findings reveal that SNR increases with magnetic field strength, T2* values differ significantly between 1.5T and 3.0T, and relaxation times decrease as magnetic field strength increases. The study suggests that MRI relaxation times, influenced by the fat/water ratio, could be potential indicators of bone quality, particularly important in diagnosing and assessing osteoporosis. These insights advocate for further development of quantitative MRI diagnostics, especially at lower field strengths, enhancing accessibility and cost-effectiveness in clinical settings.
The paper is interesting and generally well-written but will require minor corrections and additions before proceeding further. I have included detailed comments below.
Minor comments:
It seems to me that giving email addresses of all authors does not make sense, please leave only the addresses of correspondent authors.
Please supplement the introduction with more current literature, a large part of the items cited in the paper are more than 10 years old. It would be worthwhile to add some items on applications AND limitations of MRI from recent years. Please cite, for example: https://doi.org/10.3390/app11041552
https://doi.org/10.1002/jmri.28689
DOI 10.1088/1742-6596/1736/1/012028
https://doi.org/10.1016/j.ejro.2021.100342
DOI 10.1088/1742-6596/1736/1/012027
Please add details of the bioethics committee's approval i.e. the approving institution and the approval number.
Figure 3 is of poor quality and in my opinion is currently unreadable, please replace it with a better quality figure and adapt it to the requirements of the journal.
As I mentioned earlier, there are a lot of positions in the literature that are older than 10 years, I don't see the need to remove them, but it would be worthwhile to supplement them with current works from the last 5 years.
In conclusion, this paper provides valuable insights into the characterization of spongy bone MR imaging at varying magnetic field strengths. It significantly contributes to the understanding of how different MRI parameters can be optimized to improve the assessment of bone health, particularly in women, and highlights the potential for expanding the use of low-field MRI in clinical practice.
Author Response
Q. It seems to me that giving email addresses of all authors does not make sense, please leave only the addresses of correspondent authors.
R. Sorry, but these are the rules of the journal
Q. Please supplement the introduction with more current literature, a large part of the items cited in the paper are more than 10 years old. It would be worthwhile to add some items on applications AND limitations of MRI from recent years. Please cite, for example: https://doi.org/10.3390/app11041552
https://doi.org/10.1002/jmri.28689
DOI 10.1088/1742-6596/1736/1/012028
https://doi.org/10.1016/j.ejro.2021.100342
DOI 10.1088/1742-6596/1736/1/012027
R. We thank the reviewer for the suggested references that we have now inserted ( see line 43, and references from 6. to 9). Regarding the diagnosis of osteoporosis, in the last 10 years, no new methods have been developed or in any case, there have been no major advances and, therefore, the cited paper are the most important in this field. However, we have cited a more recent work (reference 23) where fatty acids and blood metabolites are quantified in healthy and osteoporotic subjects. In the paper, subjects with higher LDL are osteoporotic. We believe that low magnetic field technology will enable the further development of NMR diagnostics of osteoporosis
Q. Please add details of the bioethics committee's approval i.e. the approving institution and the approval number.
R. The study was approved by the local Ethics Committee , Fondazione Santa Lucia, Rome, Italy, with the number Prot. CE/2023_024). We have inserted this information at lines 110, 111 of the manuscript.
Q. Figure 3 is of poor quality and in my opinion is currently unreadable, please replace it with a better quality figure and adapt it to the requirements of the journal.
R. Thanks for the suggestion, we have modified figure 3 to make it more readable
Q. As I mentioned earlier, there are a lot of positions in the literature that are older than 10 years, I don't see the need to remove them, but it would be worthwhile to supplement them with current works from the last 5 years.
R. We thank the reviewer for this question. As mentioned before, the fundamental works for the development of the diagnosis of osteoporosis with NMR are not recent, that is, in the last 10 years further investigations have been carried out based on the original ones developed more than 10 years ago, but apart from some works of 2019 (reference 23 which we have now inserted) and 2022 (Mattioli et al. BONE 2022) which provide further information compared to the literature of 10 years ago, nothing else can be found. This is certainly related to the halt that this line of research has had due to the high cost of the NMR exam compared to the DXA exam. For this reason, we are convinced that the development of low-field and therefore low-cost technology will have a positive effect on the further development of osteoporosis diagnostics.
Q. In conclusion, this paper provides valuable insights into the characterization of spongy bone MR imaging at varying magnetic field strengths. It significantly contributes to the understanding of how different MRI parameters can be optimized to improve the assessment of bone health, particularly in women, and highlights the potential for expanding the use of low-field MRI in clinical practice.
R. We thank the reviewer for appreciating the manuscript and the idea of developing low-field osteoporosis diagnostics
All the changes are highlighted in yellow in the manuscript